# Metabolomics Analytics Workflow for Epidemiological Research: Perspectives from the Consortium of Metabolomics Studies (COMETS) [note 1]

**DOI:** 10.3390/metabo9070145

**Published:** 2019-07-17

**Authors:** Mary C. Playdon, Amit D. Joshi, Fred K. Tabung, Susan Cheng, Mir Henglin, Andy Kim, Tengda Lin, Eline H. van Roekel, Jiaqi Huang, Jan Krumsiek, Ying Wang, Ewy Mathé, Marinella Temprosa, Steven Moore, Bo Chawes, A. Heather Eliassen, Andrea Gsur, Marc J. Gunter, Sei Harada, Claudia Langenberg, Matej Oresic, Wei Perng, Wei Jie Seow, Oana A. Zeleznik

**Affiliations:** 1Department of Nutrition and Integrative Physiology, College of Health, University of Utah, Salt Lake City, UT 84112, USA; 2Division of Cancer Population Sciences, Huntsman Cancer Institute, Salt Lake City, UT 84112, USA; 3Clinical and Translational Epidemiology Unit, Mongan Institute, Massachusetts General Hospital, Boston, MA 02114, USA; 4Division of Gastroenterology, Department of Medicine, Massachusetts General Hospital, Boston, MA 02114, USA; 5Program in Genetic Epidemiology and Statistical Genetics, Harvard T. H. Chan School of Public Health, Boston, MA 02115, USA; 6Division of Medical Oncology, Department of Internal Medicine, The Ohio State University College of Medicine, Columbus, OH 43210, USA; 7The Ohio State University Comprehensive Cancer Center, Arthur G. James Cancer Hospital and Richard J. Solove Research Institute, Columbus, OH 43210, USA; 8Division of Epidemiology, The Ohio State University College of Public Health, Columbus, OH 43210, USA; 9Smidt Heart Institute, Cedars-Sinai Medical Center, Los Angeles, CA 90048, USA; 10Cardiovascular Division, Brigham and Women’s Hospital, Boston, MA 02115, USA; 11Department of Population Health Sciences, School of Medicine, University of Utah, Salt Lake City, UT 84112, USA; 12Department of Epidemiology, GROW School for Oncology and Developmental Biology, Maastricht University, 6200 MD Maastricht, The Netherlands; 13Division of Cancer Epidemiology and Genetics, Metabolic Epidemiology Branch, National Cancer Institute, Rockville, MD 20850, USA; 14Institute for Computational Biomedicine, Englander Institute for Precision Medicine, Department of Physiology and Biophysics, Weill Cornell Medicine, New York, NY 10021, USA; 15Behavioral and Epidemiology Research Group, American Cancer Society, Atlanta, GA 30303, USA; 16College of Medicine, Department of Biomedical Informatics, The Ohio State University, Columbus, OH 43210, USA; 17Department of Epidemiology and Biostatistics, Milken Institute School of Public Health, George Washington University, Washington, DC 20052, USA; 18COPSAC, Copenhagen Prospective Studies on Asthma in Childhood, Herlev and Gentofte Hospital, University of Copenhagen, 1165 Copenhagen, Denmark; 19Channing Division of Network Medicine, Department of Medicine, Brigham and Women’s Hospital and Harvard Medical School, Boston, MA 02115, USA; 20Department of Epidemiology, Harvard T.H. Chan School of Public Health, Boston, MA 02115, USA; 21Institute of Cancer Research, Department of Medicine, Medical University of Vienna, 1090 Vienna, Austria; 22Section of Nutrition and Metabolism, International Agency for Research on Cancer, World Health Organization, 69008 Lyon, France; 23Department of Preventive Medicine and Public Health, Keio University School of Medicine, Tokyo 160-8582, Japan; 24MRC Epidemiology Unit, Public Health, University of Cambridge, Cambridge CB2 1 TN, UK; 25The Francis Crick Institute, London NW1 1ST, UK; 26Turku Centre for Biotechnology, University of Turku, 20500 Turku, Finland; 27School of Medical Sciences, Örebro University, 702 81 Örebro, Sweden; 28Department of Epidemiology, Colorado School of Public Health, University of Colorado Denver, Anschutz Medical Campus, Aurora, CO 80045, USA; 29Life course epidemiology of adiposity and diabetes (LEAD) Center, University of Colorado Denver, Anschutz Medical Campus, Aurora, CO 80045, USA; 30Saw Swee Hock School of Public Health, National University of Singapore and National University Health System, Singapore 117549, Singapore; 31Yong Loo Lin School of Medicine, National University of Singapore and National University Health System, Singapore 119228, Singapore

**Keywords:** metabolomics, epidemiology, statistical analysis, reporting, analytical methods, data analysis, pre-processing

## Abstract

The application of metabolomics technology to epidemiological studies is emerging as a new approach to elucidate disease etiology and for biomarker discovery. However, analysis of metabolomics data is complex and there is an urgent need for the standardization of analysis workflow and reporting of study findings. To inform the development of such guidelines, we conducted a survey of 47 cohort representatives from the Consortium of Metabolomics Studies (COMETS) to gain insights into the current strategies and procedures used for analyzing metabolomics data in epidemiological studies worldwide. The results indicated a variety of applied analytical strategies, from biospecimen and data pre-processing and quality control to statistical analysis and reporting of study findings. These strategies included methods commonly used within the metabolomics community and applied in epidemiological research, as well as novel approaches to pre-processing pipelines and data analysis. To help with these discrepancies, we propose use of open-source initiatives such as the online web-based tool COMETS Analytics, which includes helpful tools to guide analytical workflow and the standardized reporting of findings from metabolomics analyses within epidemiological studies. Ultimately, this will improve the quality of statistical analyses, research findings, and study reproducibility.

## 1. Introduction

Recent advances in high-throughput methods to characterize the human metabolome present an unprecedented opportunity to strengthen epidemiological research and broaden its scope. Metabolomics is being utilized to shed light on disease etiology through objective biomarkers of exposures that are otherwise fraught with measurement error; to refine or complement our current methods of phenotypic assessment; to understand biological pathways linking exposures to health outcomes; identify early onset disease; and subtype diseases with heterogeneous etiologies [1]. Metabolomics is the comprehensive characterization of small molecules present in biospecimens such as plasma, urine, and stool. Since these small molecules reflect influences from environmental factors, as well as endogenous factors such as genetics, epigenetics, transcription, protein structure and function, and gut microbiota, metabolomics has the potential to provide a more nuanced assessment of physiology (or pathophysiology) that is often unachievable with traditional epidemiological approaches such as evaluation of single biomarkers, or self-reported data collected via questionnaires. Diverse research fields including epidemiology, systems biology, biochemistry, microbiology, pharmacology, toxicology, clinical science, and biostatistics converge through metabolomics to advance a multidisciplinary understanding of health and disease [2]. 

To date, metabolomics has shown success in screening newborns for inborn errors of metabolism, for identifying candidate biomarkers for early disease detection, particularly for some diseases like diabetes and cancer [3,4], for understanding disease mechanisms [5], and has been used to develop better measures of disease risk factors like smoking, diet, and obesity [6,7,8]. As this emerging technology is increasingly incorporated into disease research, including epidemiological studies, a bottleneck to advancing the field is the complexity and lack of standard protocols or best practices for analyzing metabolomics data [9]. Indeed, metabolomics has a unique data structure that depends on the platform (e.g., *targeted* quantification of defined groups of chemically characterized and biochemically annotated metabolites or *untargeted* semi-quantified analyses of all measurable analytes), which determines the analytical strategy to be taken. Challenges in the analysis of metabolomics data are multi-fold [10], including workflow choices for data harmonization, pre-processing (alignment, filtering), metabolite identification/annotation, data preparation (centering, scaling, and transformation) [11], imputation, and statistical approaches [12]. Moreover, since there is a high degree of collinearity between metabolites according to biochemical pathway, considering the pattern of metabolite values in addition to individual metabolites can create additional statistical obstacles. Metabolomics data management and data analysis consist of a series of complex steps that can be performed in many ways with no defined order, and some of these are optional depending on the study aims. This complexity is further compounded by the lack of adherence to a set of standard reporting guidelines [13], which makes it difficult to determine common or best practices, and leads to problems in replicating results, comparing findings, and conducting systematic reviews and meta-analyses. The purpose of this study was to summarize current practices of investigators participating in the international Consortium of Metabolomics Studies (COMETS) [14] in the analysis of metabolomics data from epidemiological studies.

## 2. Results

### 2.1. Response Rate

Thirty-three out of 47 (70%) of the participating COMETS cohorts responded to an online questionnaire up to October 2018 (See Appendix A for a summary of all results). The questionnaire inquired about current practices in the preparation, analysis, and reporting of metabolomics data. The total number of respondents was used as the denominator in calculating response rate (%). Respondents could check multiple options for each question (all that apply), and follow-up questions were asked based on some responses. When questions were unanswered, the response was denoted as “missing”. Most respondents were Principal Investigators (42%) followed by postdoctoral fellows (16%), research analysts (13%), research scientists (8%), biostatisticians (5%), and PhD students (3%). Respondents reported having conducted a median of 6 (range 1–30) analyses of metabolomics data. Multiple analyses were conducted on the same datasets, with different analysis goals.

### 2.2. Datasets

Metabolomics data were derived predominantly from untargeted metabolomics platforms (55%), with 27% derived from combining untargeted and targeted platforms and 18% from targeted only. The analysis goals were largely for biomarker discovery (91%) and/or to investigate disease etiology (82%), methodology development (31%), and other purposes (18%; e.g., metabolome-wide association study) (Figure 1a). Different study designs were used to generate metabolomics data, including: 17 cohorts, 15 nested case-control studies, 8 case-control studies, 8 case-cohorts, 5 randomized trials, and 4 cross-sectional analyses within prospective studies, with an average of 3302 participants (standard deviation 3972) (Figure 1b). Reported outcomes being analyzed included cancer (39%), cardiovascular disease (CVD) (30%), diabetes (21%), and pregnancy outcomes (6%). Human-immunodeficiency virus (HIV) infection, cardiometabolic measures, asthma, and amyotrophic lateral sclerosis (ALS) were each reported as outcomes by one respondent.

### 2.3. Power Calculations

Approximately 45% of respondents anticipated power or sample size prior to analysis. Of those that did sample size or power calculations, most were performed in Quanto (n = 6) or R (n = 4). Other resources used were Power V3 (n = 2), PASS (n = 1), and GPower (n = 1).

### 2.4. Outliers and Technical Variability

Extreme metabolite values (i.e., outliers) were evaluated by 39% of respondents, predominantly by using principal component analysis (PCA; n = 13). A subset used principal component partial R-square (PC-PR2) (n = 2) or analysis of variance (ANOVA) (n = 2) to identify outliers.

Of those that evaluated sources of metabolite variability, reported sources included batch effects (n = 6), run order (n = 3), plate, time to sample collection, and time from sample collection to freezing (all n = 1) (Figure 2a). However, most respondents did not exclude metabolites based on these sources of variability. Ten respondents reported adjusting for batch-to-batch variability (Figure 2b). Methods included adding case-control sets to each batch (n = 3), adjusting for batch in analysis (n = 4), standardizing metabolites to the batch median (n = 2), or using the Rosner approach [15] (n = 1).

Nineteen respondents measured platform reliability using coefficient of variation (CV) (n = 17) and/or intraclass correlation coefficient (ICC) (n = 12) (Figure 2c). The range of CVs reported for completed studies was 0–78% (most up to 20%), while the range of ICCs reported was up to 1.0 (most > 0.90). Seven respondents reported using this information to exclude metabolites from analysis, but criteria for exclusion were variable (e.g., CV > 20% or ICC < 0.40) (Figure 2d).

### 2.5. Data Preparation

Centering, scaling, and data transformation are data preparation methods used in metabolomics studies [11]. Thirty-nine percent of respondents reported centering individual metabolite values while 42% did not (15% missing). The most common method was centering to the mean (n = 10). Scaling methods included Pareto-scaling (n = 2), auto-scaling/z-transformation/standard-deviation scaling (n = 7), probit-score scaling (n = 1), and median absolute deviation (MAD) (n = 1). Most respondents reported transforming metabolite data (85%; 15% missing), including by log-transformation (n = 21).

### 2.6. Missing Data

Most respondents (85%) reported that their metabolomics data had missing values. Missingness was due to the limit of detection (LOD)/quantification of the platform (n = 24), low abundance (n = 13), and rare metabolites (n = 8). Co-elution issues and failed quality control (QC) were also reported (n = 1 each) (Figure 3a).

Twenty-one respondents reported imputing the missing values while seven did not (n = 5 did not answer). The most common approach to treat missing values was to replace these by a fraction of the lowest value (n = 15). Replacing missing values by zero or by the minimum value were each reported by three respondents. K-nearest neighbor imputation (KNN) [16] was used by one respondent while none used multiple imputation by chained equations (MICE) or Markov chain Monte Carlo (MCMC) (Figure 3b). Most respondents (n = 15) reported excluding metabolites with a percent of missingness above a certain threshold (median 50%; range 5% to 90%). Dichotomization, categorization as missing or min-median or median-max, imputation to the mean, flagging, and complete exclusion were each reported by one respondent.

### 2.7. Statistical Analysis Methods

Respondents used multiple statistical analysis strategies to analyze metabolomics data (Figure 4). The most common strategy was univariate regression (e.g., linear regression of a single exposure on a single metabolite) (76%) followed by multiple/multivariable analysis of metabolites on the exposure of interest, with adjustment for covariates (67%), principal component analysis (PCA, 61%), metabolome-wide association study (MWAS, 48%), partial least squares-discriminant analysis (PLS-DA, 42%), partial correlation (30%), partial least squares analysis (PLS, 18%), and hierarchical clustering (18%). The following analysis techniques were each reported by one respondent: canonical correspondence analysis (CCA), treelet transform, K-means clustering, least absolute shrinkage and selection operator regression (LASSO), supervised gradient descent, random forest, support vector machines (SVM), weighted gene co-expression network analysis (WGCNA), metabolite set enrichment analysis, over-representation analysis, differential networks, hierarchical cluster analysis, Bayesian non-parametric methods, orthogonal projections to latent structures discriminant analysis (OPLS-DA) and generalized linear mixed models (GLM). A third of respondents reported having used variable selection methods incorporating penalization, including LASSO, SVM, and sparse seemingly unrelated regression (SUR). Mediation analysis was conducted by 15% of respondents.

Almost half of the respondents (48%) assessed the performance of biomarker classification using area under the receiver operator characteristic curve (AUC) (n = 16), net reclassification improvement (n = 2), sensitivity/specificity/positive predictive value/ negative predictive value (n = 1) and PLS-DA (n = 1).

Approximately 40% of respondents measured metabolite intercorrelations. The most common method for assessing metabolite intercorrelations was partial correlation (n = 11). Gaussian graphical modeling (GGM; n = 2) and WGCNA (n = 1) were also used for this purpose.

A quarter of respondents conducted network analyses. WGCNA (n = 3) and unspecified methods incorporated into programs within the analytic resource MetaboAnalyst (n = 2) were the most common approaches followed by GGM with linkage to biological pathway, BayesNet, Gene-Set Enrichment Analysis (GSEA), over representation analysis (ORA), Metscape, and yED graphical networking software (n = 1 each).

### 2.8. Cross Validation and External Validation

Seven respondents (21%) performed cross-validation (CV) analysis. Six respondents used k-fold CV and one reported simulation/permutation of data. The proportion of training data varied (60% to 90%) as did the proportion of testing data (10% to 40%). Five respondents used bootstrapping, and 11 externally validated findings in another cohort.

### 2.9. Visualization

Respondents visualized results using heat maps (n = 17), volcano plots (n = 6), Manhattan plots (n = 5), forest plots (n = 2), and individual approaches (n = 1).

### 2.10. Multiple Testing Correction

Correcting for testing multiple hypotheses following analysis of metabolomics data (e.g., when regressing multiple metabolites on an exposure, separately) was done by 79% of respondents (Figure 5). The Benjamini-Hochberg false discovery rate (FDR) was the most common approach (n = 22), followed by Bonferroni correction (n = 12), Bonferroni-Holm, Dunn-Sidak, and permutation tests (n = 1 each).

### 2.11. Meta-Data

In total, 9 of 33 respondents (27%) appended metabolite meta-data for publication (Figure 6). Unique identifiers such as those from publicly available metabolomics databases such as Human Metabolome Database (HMDB) and PUBCHEM were most often appended to metabolomics study results (n = 5). The addition of pathway information and mass-to-charge ratio (m/z ratio) were also reported (n = 4) along with retention time (n = 3) and internal database compound tracking number in the platform’s chemical library (COMP ID) (n = 1).

### 2.12. Annotations

Eight respondents (24%) annotated metabolites. Of those, databases that were used for annotation included HMDB (n = 6), Metlin (n = 3), KEGG pathways (n = 2), Metabolon proprietary system (n = 1), and the National Institute for Standards and Technology (NIST) library (n = 1) (Figure 7).

### 2.13. Coding Language

Most respondents (61%) wrote their statistical code in R (Figure 8). Other popular coding languages included SAS (33%) and STATA (15%). Fewer respondents used other languages (e.g., Python, Haskell, Matlab; n = 1 each). Four respondents (19%) used a statistical coding style guide (e.g., tidyverse, Google’s R style guide) in the design of their code.

### 2.14. Software

Of the numerous open source software packages available online (for examples, see Table 1), none were leveraged by respondents to this survey. Rather, they reported writing original code or using R packages for analysis of metabolomics data.

### 2.15. Minimum Reporting Standards

There were many suggested minimum reporting standards for analysis of metabolomics data, including reporting: study aim and objectives; study hypothesis; statistical assumptions; overall analytical strategy; metabolomics data standard operating procedures; pre-analytical processing measures; analytical platform; quality control measures used; statistical packages and software used; strategy for adjustment for multiple comparisons; meta-data; effect estimates and confidence (betas, confidence intervals, P-values and Q-values for both nominal and statistically adjusted results); confounder selection; cross-validation strategy; external validation strategy; and providing statistical code for replication.

## 3. Discussion

The application of metabolomics in epidemiological studies has increased dramatically in recent years [17]. The COMETS consortium is currently collaborating on large-scale replication studies and meta-analyses of metabolomics data [14]. With an overall study population in excess of 130,000 participants, median age 51 years (range 0–100 years) representing European, Asian, African, Hispanic, native Hawaiian and other mixed populations, COMETS is a rich resource for addressing research questions that leverage blood metabolomics data. Given that application of metabolomics technology to epidemiological studies is an emerging field, we conducted a survey of participating COMETS cohort representatives to gain insights into the strategies and procedures used for analyzing metabolomics data, including data pre-processing, analysis, and reporting of results. With a range of experience levels analyzing metabolomics data, participating COMETS cohorts have analyzed and published on both targeted and untargeted metabolomics data from a variety of study designs in relation to disease risk factors [8,18,19,20,21,22,23,24,25,26,27] and many disease outcomes, predominantly cancer, CVD, and diabetes [28,29,30,31,32,33,34,35,36,37,38,39,40]. However, there was little consensus on approaches to data pre-processing, statistical analysis or reporting of results, which is echoed in the broader metabolomics community [13].

Metabolomics studies have predominantly investigated disease etiology or biological mechanisms underlying progression and for biomarker discovery, using metabolomics data generated on both targeted or untargeted platforms in the context of a variety of epidemiological study designs [23,40,41,42,43]. The type of study design selected is typically related to the research question of interest in addition to the availability of metabolomics data and the frequency of the outcomes of interest in that same study. Importantly, statistical analytical considerations will differ depending on study design. For instance, a nested case-control study may require propensity-score matching to avoid ascertainment bias, by controlling for the probability to be included in the metabolomics data based on eligibility criteria [38]. By contrast, large cohort studies that investigate rarer outcomes may utilize multivariable regression or tree-based analysis. Furthermore, data reduction approaches may be needed for high-dimensional untargeted metabolomics data.

### 3.1. Data Pre-Processing

A first step in analysis of metabolomics data includes data pre-processing. Extreme metabolite values are frequently observed even after applying any one of a variety of normalization methods to large metabolomics datasets [10], yet less than half of respondents evaluated them. A variety of sources drive extreme metabolite values including pre-analytical conditions (e.g., processing delay), batch variability (due to a variety of technical factors), misalignment (between or within batches), chemical instability (typically manifesting as within-batch variation), true biological variation that may arise from rare genetic determinants or rare exposures, other random effects that are not easily classifiable, or some combination of the above [44]. Filtering or censoring extreme values can reduce the skewness of a distribution and stabilize metabolite variance, improving reliability and interpretability of statistical analysis results. However, this approach could lead to misclassification or loss of information, particularly for metabolites that may represent rare exposures that could be associated with rare outcomes (e.g., certain drugs, chemicals, or foods) or extreme manifestations of common outcomes. Alternately, transformations based on rank only, such as the probit transformation, represent an elegant solution and avoid such exclusions. Log scaling and winsorization are other strategies to reduce the influence of outliers.

Missingness among metabolite data was also a common occurrence. Missing values may be due to biological factors, such as metabolites being absent (e.g., drug metabolites), and various technical limitations in computational detection, including separation of metabolite signal to noise, low signal intensity (e.g., lower value for detection), and measurement error [45]. Various analytical approaches for imputation were reported, including replacement with zero, half (or another proportion) of the minimal detected value, or more complex statistical approaches such as KNN [16], PCA, or random forest imputation [45]. Missing values are often assumed to be due to technical limitations including being below the metabolomics platform’s LOD; however, truly absent values must be considered. It is challenging to distinguish between the two as this task requires extensive biochemical knowledge. However, imputing missing values when the metabolite is absent (e.g., for a drug metabolite), is not meaningful and may result in spurious results. Another important consideration is the percentage of missing values per metabolite. Imputation for metabolites with a large proportion of missing values may result in a metabolite with low information content but increasing the multiple testing burden. In this case, exclusion of the metabolite or dichotomization to missing/not-missing values may represent more suitable alternatives. Prior to removing such metabolites, evaluating their relationship with the experimental condition or exposure of interest should still be considered to ensure that valuable information is not discarded. Therefore, analysts must consider the chemical nature or source of extreme or missing metabolite values in determining how to deal with them.

Batch variability or signal drift adjustments were not commonly conducted. These are considered a standard part of the workflow in the field. It is likely that the lack of reporting batch adjustments was due to the use of commercial platforms (e.g., Metabolon Inc. [46] and the Broad Institute [47]) that conduct batch and other laboratory adjustments as part of their standard operating procedures. In some cases when data pre-processing is done by the metabolomics laboratory (or bioinformaticians who work closely with the laboratory), the steps used are not always available to the end user. Additionally, depending on the type of analytical instrument used (i.e., NMR versus MS-based), pre-processing steps could be drastically different [48]. We found that centering, scaling, and transformation of metabolomics data were common, such as adjustment of feature/metabolite intensity by the median across samples, or standard normalization approaches like log_2_ or log_10_ transformation [11,49,50,51]. The most appropriate normalization method for any given large cohort experiment depends on the type of mass spectrometry method used and the size of the cohort experiment; ongoing work is being done to investigate the relative performance of different normalization methods applied to large cohort metabolomics data [52,53].

Most respondents (58%) calculated reliability measures such as metabolite CVs and ICCs, although they generally did not exclude metabolites using these criteria. Encouragingly, of those that measured ICCs, reliability was excellent, on average (ICC>0.9). The variability of metabolite levels in population studies is an important consideration when estimating study power and the true compared with observed study effect estimates. Three main sources of variability include (1) between-subject variability or usual level in the population, (2) within-subject variability representing the usual level within an individual (e.g., year-to-year variability), and (3) technical or laboratory reproducibility or variance expected from identical samples. These components can be integrated into the technical ICC or the proportion of the total variation that is attributable to biologic variance versus random laboratory error [54]. Studies with higher biologic variance and lower technical metabolite variance (e.g., cohorts enriched with certain disease risk traits and measures of primarily highly abundant metabolites) may have higher study power to detect epidemiological associations. Repeated samples can also reduce within-individual variability and improve study power [54,55].

### 3.2. Data Analysis

#### 3.2.1. Analytic Approaches

Following multivariate dimension reduction and/or identification of relevant metabolites, epidemiological methods such as multivariable regression analysis were commonly used for data analysis, but novel and more complex approaches such as adaptations of penalized regression and network analysis are also emerging [56,57,58]. While the data pre-processing pipeline should be consistent, the data analysis techniques used are driven by the goals of the study. For instance, predictive models like LASSO may be useful for biomarker discovery, but methods applied to the study of mechanisms or metabolomic profiles associated with exposures will depend on the directionality of the underlying biology. Our findings are consistent with a recent survey of the broader metabolomics community that surveyed metabolomics workflow and computation strategies [59]. As an emerging field, metabolomics databases are still incomplete and thus interpreting results from metabolomics datasets in a biological context is challenging. Data-driven network-based approaches support a better understanding of the biological processes driving exposure-disease associations [60], and can provide biological information independent of background databases as well as incorporating unknown metabolites [61]. Examples include Gaussian graphical modeling [62], weighted gene co-expression network analysis [63], sparse network modeling [64], Bayesian approaches [65], and machine learning methods such as random forests [66].

Automated text mining is a bioinformatics approach and queries databases to provide biological context based on a metabolite list [61]. As metabolomics technologies continue to evolve and expand to include larger numbers of novel (i.e., unknown) molecules, the ability for existing databases to provide structure for network analyses becomes more limited. For chemically characterized metabolites, chemical pathway analysis [67] can identify biologically meaningful metabolite groups (i.e., representing biochemical pathways) using information from biochemical databases, and may also serve to strengthen power to detect associations compared with evaluating single metabolites.

To facilitate metabolomics analyses, which are considerably more complex than traditional epidemiological studies, online platforms that aid in identification of relevant biochemical pathways, such as Metaboanalyst [68], the Metabolomics Workbench [69], and others [70] were developed. Moreover, an increasing number of studies are measuring more than one ‘omics data type [71,72] and methods that integrate multiple ‘omics datasets are also under development. Examples of useful tools available for processing and analysis of metabolomics data are presented in Table 1.

#### 3.2.2. Correction for Multiple Statistical Testing

One of the most important differences between conducting a single biomarker versus a metabolomics analysis within an epidemiological study is the number of tested hypotheses. In order to account for the high number of study hypotheses in many metabolomics studies (particularly for untargeted metabolomics), several methods are available to reduce the rate of Type I errors. Respondents predominantly used the false discovery rate (FDR), which is considered a less stringent approach, followed by Bonferroni correction to account for testing multiple hypotheses. These reflect the most widely used methods currently, although FDR approaches that account for highly correlated data are lacking. A detailed description of these approaches together with proposed alternatives, such as resampling-based strategies, have been reported elsewhere [81]. There is a need to determine the most appropriate method for correcting for multiple statistical testing given the correlated nature of metabolomics data.

#### 3.2.3. Classification Performance

Reporting of classification performance is an important step in translating risk factor or disease biomarkers to a clinical setting. Approximately half of respondents noted conducting such analyses. Inconsistent reporting of biomarker classification performance and poor transparency in reporting prediction algorithms have been observed among the broader metabolomics community [82]. Biomarker discovery includes selecting biomarkers that maximally discriminate cases from controls, validating the biomarker panel, and deriving a final model with a fixed mathematical algorithm for predicting the clinical outcome [82]. Measures of biomarker sensitivity, specificity, and receiver operator characteristic (ROC) curves are used to assess the performance of biomarkers for classifying disease diagnosis, prognosis, and risk factor or prediction biomarkers [83]. For disease biomarkers, reporting of ROC curves for disease classification would support biomarker comparison across studies.

#### 3.2.4. Meta-Data

Metabolite metadata includes information on metabolomics platform and procedures such as software used, reliability (CV, ICC), chemical identification, mass-to-charge ratio and retention time, chemical pathway, and biological information, among others. This information was not commonly presented in publications. Metadata is crucial for linkage across metabolite databases to retrieve metabolite information, conducting between-study comparisons, metabolite annotation, and informing replication efforts. Databases such as the Human Metabolome database (HMDB) [84] and PubChem [85] assign unique identifiers and compile useful accompanying metadata from previous studies. Quality of metabolomics metadata has been reviewed previously [86]. In an epidemiological setting, appending metabolite metadata will support future replication and meta-analysis efforts, such as those proposed within COMETS.

#### 3.2.5. Validation

Cross- and external validation of metabolomics analyses were uncommon among COMETS respondents. External validation represents the gold-standard approach to show generalizability of results. The lack of independent validation is a major challenge in metabolomics biomarker discovery [87]. Validation is particularly important when not all metabolites are stable over long periods of time and across sample collection methods [88]. Extensive costs of acquiring metabolomics data and difficulties in obtaining suitable samples (e.g., in case of a rare disease) may complicate external validation. In that case, as an alternative one may choose to apply cross-validation [89] or double cross-validation [90] by conducting the main analysis on a subset of the study participants and validating the results on the remaining subset. This may often represent the only way to validate results, but the investigators must keep in mind that this approach leads to lower power in both the discovery and validation datasets. COMETS provides a unique opportunity for increased biomarker validation in cohorts with diverse participant demographics and clinical features.

#### 3.2.6. Coding Language

Most statistical programming was conducted in R. There are many freely available software packages for metabolomics analysis, summarized by the Metabolomics Society [91] and elsewhere [92]. Moreover, platforms such as Galaxy, originally designed for developing genomics research workflow pipelines, have been applied to metabolomics [59]. The recognized advantages of using R and other open source coding languages include the open source framework, which allow investigators and programmers to fully access each line of code and edit or adapt code as needed for a given data management or analysis purpose. For this reason, flexible and open tools are likely to continue being developed in R, with more user-friendly adaptations of code being developed as ready-to-use R packages. Standardizing statistical analysis approaches in workflows developed as R packages will also help to augment potential for replication of analyses across large cohort study designs.

### 3.3. Reporting of Data Analysis Workflow

Pre-analytical and analytical strategies are often poorly reported within scientific manuscripts [93]. A recent review of 27 studies published between 2008 and 2014 assessed the standard of reporting of data management and analysis steps in metabolomics biomarker discovery studies and investigated whether the level of detail reported allows basic understanding of the steps employed and/or reuse of the protocol [13]. The authors concluded that there is unclear and incomplete reporting of these procedures in metabolomics studies that preclude replication in another study. Standardized reporting of observational studies in epidemiology is outlined by the STROBE statement [94] and CONSORT statement [95], with application to genetic epidemiology studies through the Standardized Reporting of Genetic Association Studies (STEGA) [96], among others. Recommendations for standardizing reporting of epidemiological studies with metabolomic analyses and infrastructure to support it have also been proposed [59,77,97], including: experimental design; analytical dataset format; sample handling and data acquisition parameters; post-instrument data processing; multivariate statistical procedures; data modeling; and model validation. Based on our findings, a summary of observed workflow is presented in Figure 9.

In summary, there is a need to develop standardized analytical workflows, reporting standards, and tools for analysis of metabolomics data in epidemiological studies. Our survey was conducted among a small number of respondents, but they were representatives of their respective prospective cohorts and therefore represent the views of a larger population of analysts. Nonetheless, conducting similar surveys in a larger sample would strengthen the current findings. To coordinate and streamline consortium-based data analyses, COMETS developed COMETS Analytics, a secure online statistical analysis platform for metabolomics data analysis [98]. COMETS Analytics processes summary data generated by participating cohorts, as opposed to individual-level data. The web-based application performs three main tasks: it harmonizes metabolite identifiers across cohorts that utilized different metabolomics platforms, conducts statistical analyses in large batches of user-defined models (including correlation analysis and multivariable regression), and produces standardized, meta-analysis ready output. The data pre-processing steps are completed by each cohort according to their workflow prior to analysis in COMETS Analytics. The source code is publicly available through GitHub (https://rdrr.io/github/CBIIT/R-cometsAnalytics/). The platform aims to accelerate data analysis and lower error rates compared with more conventional approaches [14]. Educational tools are under development to guide analytical workflow and reporting of findings. These resources are intended to be open-source and freely available to the public to support rigorous research efforts and training in the analysis of metabolomics data.

## 4. Materials and Methods

### 4.1. Study Population

The study population included representatives of 47 prospective cohorts participating in the Consortium of Metabolomics Studies (COMETS) [14]. Representatives were cohort Principal Investigators and those with hands-on experience conducting analyses of metabolomics data from their respective cohorts (i.e., research analysts, biostatisticians, research scientists, postdocs, and graduate students).

### 4.2. Questionnaire

Participants were invited to complete an online questionnaire designed to collect information on the workflow and analytical strategies used for past and current metabolomics analyses of epidemiological data within their cohort. The questionnaire was conducted using Survey Monkey (https://www.surveymonkey.com) between 8 June and 5 October 2018. Questions were informed by common reporting of metabolomics analysis methods in the literature and were either multiple choice or multiple choice with the option of open-ended responses. Topics included: (1) Study information (purpose of the analysis, study design, type of analysis [targeted, untargeted, or both]); (2) Exploratory analyses and pre-processing (power calculations, data normalization, dealing with missing data, assessing technical platform reliability); (3) statistical analysis (analytic strategies, visualizations, cross-validation); (4) metabolite annotations; (5) other (statistical software, biostatistician input, open-source packages used, statistical coding language, minimal reporting standards). Responses were collated and summarized as frequency of responses (N/%) based on total number of respondents. Open-ended responses were summarized. The questionnaire can be found in the Appendix A.

## 5. Conclusions

We conducted a survey of 47 participating COMETS cohort representatives to gain insights into the strategies and procedures used for analyzing metabolomics data in epidemiological studies worldwide. Our results indicate a large variety of analytical strategies being applied, from data pre-processing and quality control to statistical analysis and reporting of the findings. These methods are both common epidemiological approaches and emerging novel methods. We found that there was consensus on several aspects of metabolomics analysis workflow, including data transformation/normalization, dealing with missing values, multiple testing correction, and choice of statistical software. However, more thought is merited on what would be most appropriate for metabolomics data, such as the optimal multiple testing correction given its highly correlated nature. Moreover, there is a clear need to establish benchmarks in relation to other data pre-processing steps, use of cross and external validation, and minimum reporting standards including reporting metabolite reliability estimates and appending meta-data to study results. Although there was a wide range of analytic approaches applied to metabolomics data, it is likely that analytical choices will continue to depend on the study question and the nature of the data (i.e., targeted or untargeted). Altogether, our results indicate the need for standardized analytical workflows, reporting standards, and openly shared tools for analysis of metabolomics data in large-scale epidemiological studies—an approach that has catalyzed scientific progress in other similarly expansive fields [99]. Accordingly, the open-source COMETS Analytics initiative [98] is currently developing a set of educational modules and analytic code using common statistical coding language to support the analysis and interpretation of large-scale metabolomics data derived from epidemiological studies. Our current findings can be leveraged to inform the development of minimum reporting standards for metabolomics data analysis to support best practices and study reproducibility. Ultimately, we anticipate this will improve the quality of metabolomics data analysis and results and enable a better comparison and interpretation of the results across metabolomics studies.

## Figures and Tables

**Figure 1 metabolites-09-00145-f001:**
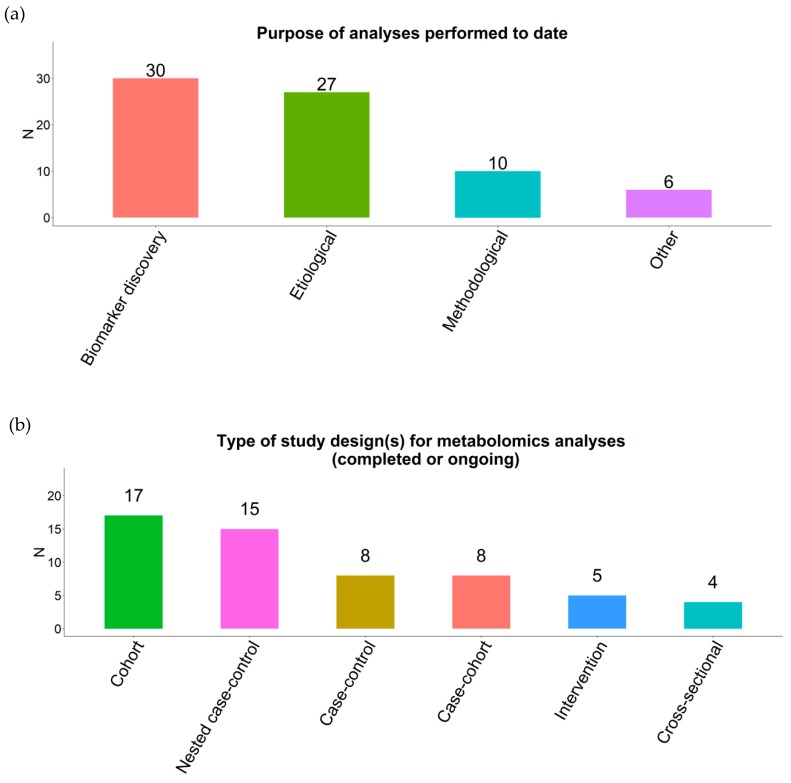
Description of study purpose (**a**) and study design (**b**) of participating Consortium of Metabolomics Studies (COMETS) cohorts.

**Figure 2 metabolites-09-00145-f002:**
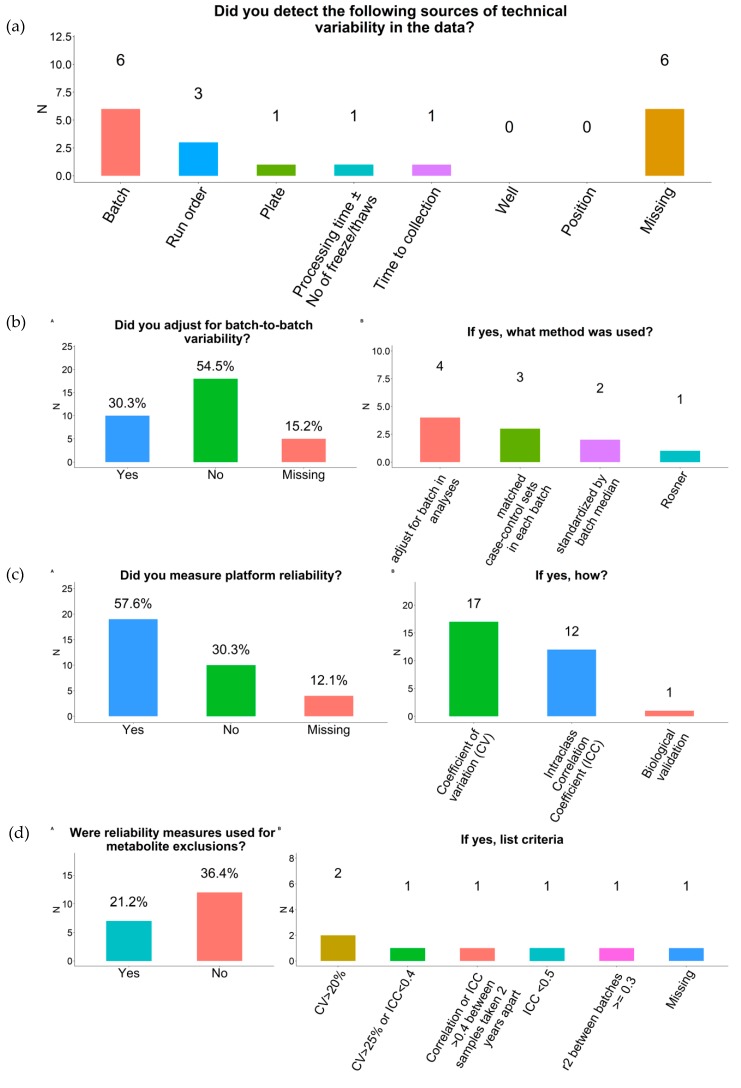
Reliability measures among participating COMETS cohorts. (**a**) Sources of technical variability; (**b**) Batch-to-batch variability; (**c**) Platform reliability; (**d**) Metabolite exclusion criteria. Missing refers to the proportion of respondents that did not answer the question.

**Figure 3 metabolites-09-00145-f003:**
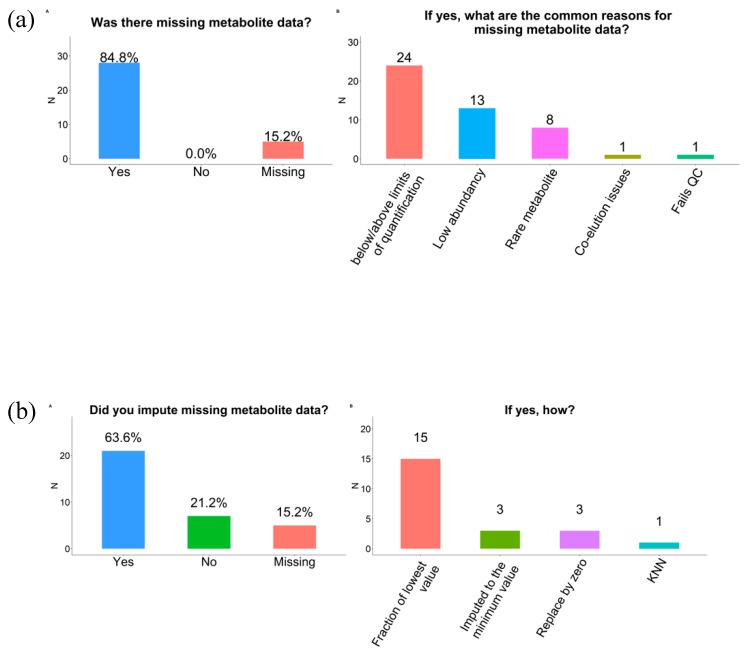
Data pre-processing steps conducted among participating COMETS cohorts. (**a**) Missingness; (**b**) Imputation of missing values.

**Figure 4 metabolites-09-00145-f004:**
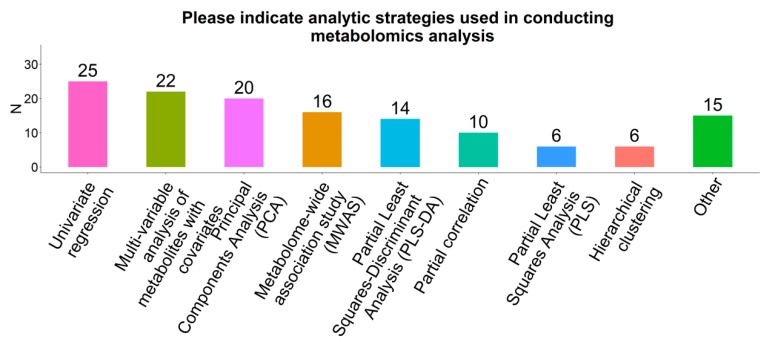
Analytic strategies employed for metabolomics data among participating COMETS cohorts.

**Figure 5 metabolites-09-00145-f005:**
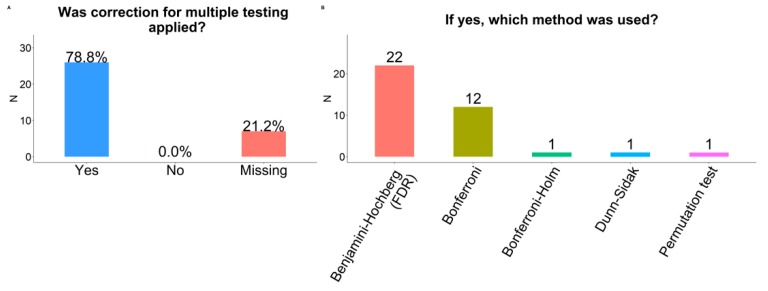
Strategies for correcting for multiple hypothesis testing among participating COMETS cohorts. (**a**) Use of multiple testing correction (yes/no); (**b**) Methods for correcting for multiple hypothesis tests.

**Figure 6 metabolites-09-00145-f006:**
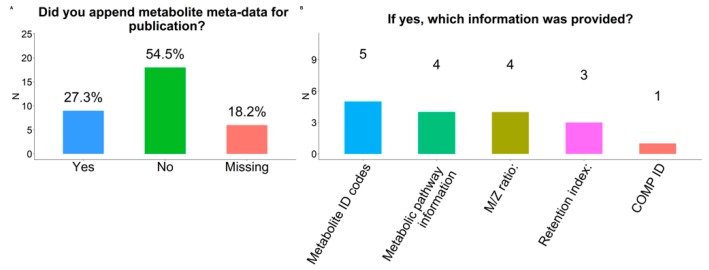
Appending metabolite meta-data in publications of findings from participating COMETS cohorts. (**a**) Include meta-data for publication (yes/no); (**b**) Information provided in appended meta-data.

**Figure 7 metabolites-09-00145-f007:**
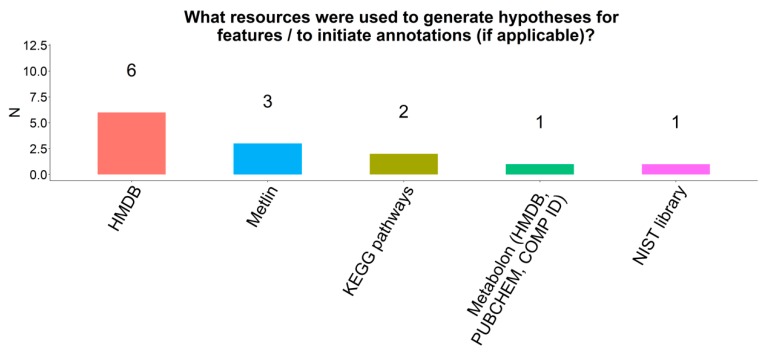
Strategies for metabolite annotation among participating COMETS cohorts.

**Figure 8 metabolites-09-00145-f008:**
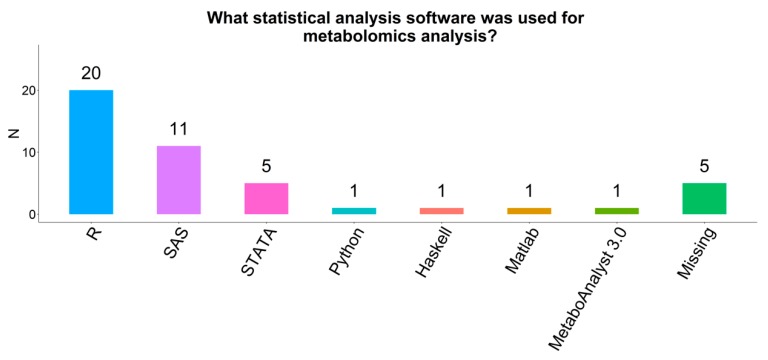
Statistical analysis software used by participating COMETS cohorts.

**Figure 9 metabolites-09-00145-f009:**
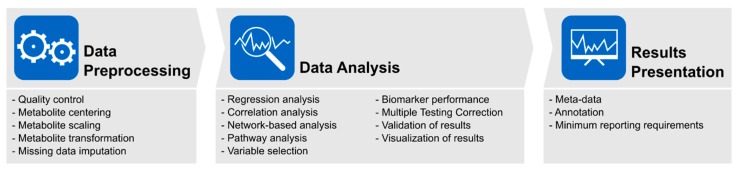
Suggested metabolomics analysis workflow.

**Table 1 metabolites-09-00145-t001:** Resources available for analysis and interpretation of metabolomics data. ^a^

Resource	Name	Description	Website
Consortia and Societies	Consortium of METabolomics Studies (COMETS)	Consortium of prospective studies with blood metabolomics data.	https://epi.grants.cancer.gov/comets/ [14]
	Metabolomics Society	Summary of metabolomics databases.	http://metabolomicssociety.org/
	COordination of Standards in MetabOlomicsS (COSMOS)	Standards for data dissemination.	http://cosmos-fp7.eu/ [73]
Statistical Analysis Tools; Meta-Data and Other Resources	Metabolomics Workbench	Metabolomics resource sponsored by the Common Fund of the National Institutes of Health.	http://www.metabolomicsworkbench.org/ [69]
	MetaboAnalyst	Program for statistical, functional and integrative analysis of metabolomics data.	https://www.metaboanalyst.ca/MetaboAnalyst/faces/home.xhtml [68]
	Metabox	A toolbox for metabolomic data analysis, interpretation, and integrative exploration.	http://kwanjeeraw.github.io/metabox/ [74]
	MZmine	A modular framework for processing, visualizing, and analyzing mass spectrometry-based molecular profile data.	http://mzmine.github.io/ [75]
	XCMSOnline	Metabolomics data processing and analysis platform.	https://xcmsonline.scripps.edu/landing_page.php?pgcontent=mainPage [76]
	Workflow4Metabolomics	Collaborative research infrastructure for computational metabolomics.	https://workflow4metabolomics.org/ [77,78]
	PhenoMeNal	Cloud-based platform for metabolomics processing and analysis.	http://phenomenal-h2020.eu/home/ [79]
	Metabolomics Tools Wiki	Classified and searchable list of metabolomics software and tools.	https://raspicer.github.io/MetabolomicsTools/
	MetaboLights	Database for metabolomics experiments and derived information.	https://www.ebi.ac.uk/metabolights/ [80]
	MetabolomeXchange	An international data aggregation and notification service for metabolomics.	http://www.metabolomexchange.org/site/

^a^ The metabolomics resources cited here are provided as a summary of existing tools rather than an endorsement of specific tools.

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
