# Peer review of "Metabolomics Analytics Workflow for Epidemiological Research: Perspectives from the Consortium of Metabolomics Studies (COMETS)†"

_metabolites, 2019, doi:10.3390/metabo9070145_

Round 1
Reviewer 1 Report
In their manuscript, Playdon et al. discuss the application of metabolomics technology in epidemiological studies and systematically review the strategies, procedures used for metabolomics data analysis based on a survey of COMETS cohort representatives. In my opinion, this manuscript is concisely written and contains interesting information related to the application of metabolomics in large-scale epidemiological studies. In particular, this study is focused on the data processing and quality control, which are the major bottlenecks of metabolomics studies. The authors discuss the current situations of metabolomics technology and make some very good proposals regarding the strategies and methods that can be used for metabolomcis studies. The only drawback of this study is that the number of participants for the survey is kind of small, which makes some of the conclusions drawn in this study not so convincing. A future survey based on larger number of participants is recommended. Overall, I found this study a stimulating and useful contribution to the metabolomcis fields, which will definitely help the improvement of the quality of data analysis and interpretation of the results. Therefore, it is adequate to be published in “Metabolites” in its current form.
Author Response
Thank you for your comments. We have included the following in the summary paragraph (line 508): “Our survey was conducted among a small number of respondents, but they were representatives of their respective prospective cohorts and therefore represent the views of a larger population of analysts. Nonetheless, conducting similar surveys in a larger sample would strengthen the current findings.”

Reviewer 2 Report
The manuscript is about results from a questionnaire sent to 47 members of COMETS to gain insights into current strategies/procedures for analyzing metabolomics data.
I think the research is original and can be of use for experimentalists analyzing metabolomics data and not in touch with biostatisticians / autodidacts. For people familiar with such analyses the manuscript may be of lesser use but still a good check up to see what is going around.
Remarks:
line 187, mean centering is in this case not connect to PCA but some other analysis?
line 213, univariate regression is t-tests/hypothesis testing or linear models in general? The label univariate indicates one response/trait/class but still multiple metabolites and covariates can be used as input.
line 253, in connection to which methods is multiple correction done?
line 286, maybe examples can be given?
line 335, for outlier detection, or rather reducing the influence, log scaling can also be mentioned as outliers become less influential after log scaling.
line 355, removing metabolites with high percentages missingness should always preceded with a correlation analysis wrt experimental/observational factor to reduce the risk of throwing away valuable metabolites.
line 359, I don't fully agree that batch/drift correction is considered pre-statistical analysis as I see this more and more being executed by statisticians. The reasons is that more and more experiments become so large that batch effects are inevitable.
general remark:
- I'm not sure if multivariable regression in this context refers to multiple regression or multivariate regression.
The use of the label 'missing' for unanswered question got a bit unclear in figure 3a and 3b
Author Response
Reviewer B
The manuscript is about results from a questionnaire sent to 47 members of COMETS to gain insights into current strategies/procedures for analyzing metabolomics data. I think the research is original and can be of use for experimentalists analyzing metabolomics data and not in touch with biostatisticians / autodidacts. For people familiar with such analyses the manuscript may be of lesser use but still a good check up to see what is going around.
Remarks:
line 187, mean centering is in this case not connect to PCA but some other analysis?
We have clarified that mean centering was conducted on individual metabolites (line 190): “Thirty-nine percent of respondents reported centering individual metabolite values while 42% did not (15% missing).”
line 213, univariate regression is t-tests/hypothesis testing or linear models in general? The label univariate indicates one response/trait/class but still multiple metabolites and covariates can be used as input.
We have clarified the meaning of univariate and multivariable regression with the following (line 216): “The most common strategy was univariate regression (e.g. linear regression of a single exposure on a single metabolite) (76%) followed by multiple/multivariable analysis of metabolites on the exposure of interest, with adjustment for covariates (67%),…”
line 253, in connection to which methods is multiple correction done?
We have provided an example to clarify multiple testing correction (line 257): “Correcting for testing multiple hypotheses following analysis of metabolomics data (e.g. when regressing multiple metabolites on an exposure, separately) was done by 79% of respondents”.
line 286, maybe examples can be given?
We refer to Table 1 to provide some examples (line 292): “Of the numerous open source software packages available online (for examples, see Table 1), none…”
line 335, for outlier detection, or rather reducing the influence, log scaling can also be mentioned as outliers become less influential after log scaling.
This is a good point. We have included log scaling, in addition to winsorization, as alternatives (line 347): “Log scaling and winsorization are other strategies to reduce the influence of outliers.”
line 355, removing metabolites with high percentages missingness should always preceded with a correlation analysis wrt experimental/observational factor to reduce the risk of throwing away valuable metabolites.
We have included this recommendation (line 363): “Prior to removing such metabolites, however, evaluating their relationship with the experimental condition or exposure of interest should still be considered to ensure that valuable information is not discarded.”
line 359, I don't fully agree that batch/drift correction is considered pre-statistical analysis as I see this more and more being executed by statisticians. The reasons is that more and more experiments become so large that batch effects are inevitable.
We have modified this sentence to emphasize batch correction as a standard part of the workflow (line 367): “These are considered a standard part of the workflow in the field.”
general remark:
- I'm not sure if multivariable regression in this context refers to multiple regression or multivariate regression.
Please see our response to the reviewer’s second comment, where we clarify multivariable regression.
The use of the label 'missing' for unanswered question got a bit unclear in figure 3a and 3b
We have included clarification of the meaning of ‘missing’ data in the results section (line 138): “When questions were unanswered, the response was denoted as “missing”.
